

# Soft Bigram distance for names matching

Mohammed Hadwan[1,2,3], Mohammed A. Al-Hagery[4], Maher Al-Sanabani[5] and Salah Al-Hagree[6]

[1] Department of Information Technology, College of Computer, Qassim University, Buraydah, Saudi Arabia
[2] Intelligent Analytics Group (IAG), College of Computer, Qassim University, Buraydah, Saudi Arabia
[3] Department of Computer Sciences, Faculty of Applied Sciences, Taiz University, Taiz, Yemen
[4] Department of Computer Science, College of Computer, Qassim University, Buraydah, Saudi Arabia
[5] Faculty of Computer Science and Information Systems, Thamar University, Thamar, Yemen
[6] Department of Computer Sciences & Information Technology, IBB University, IBB, Yemen

## ABSTRACT

**Background**. Bi-gram distance (BI-DIST) is a recent approach to measure the distance between two strings that have an important role in a wide range of applications in various areas. The importance of BI-DIST is due to its representational and computational efficiency, which has led to extensive research to further enhance its efficiency. However, developing an algorithm that can measure the distance of strings accurately and efficiently has posed a major challenge to many developers. Consequently, this research aims to design an algorithm that can match the names accurately. BI-DIST distance is considered the best orthographic measure for names identification; nevertheless, it lacks a distance scale between the name bigrams.
**Methods**. In this research, the Soft Bigram Distance (Soft-Bidist) measure is proposed. It is an extension of BI-DIST by softening the scale of comparison among the name Bigrams for improving the name matching. Different datasets are used to demonstrate the efficiency of the proposed method.
**Results**. The results show that Soft-Bidist outperforms the compared algorithms using different name matching datasets.

Corresponding author
Mohammed Hadwan,
m.hadwan@qu.edu.sa

## INTRODUCTION

Currently, name matching is one of the hottest topics in the emerging data science area, where the BI-DIST is a recent and significant approach for name matching by measuring the distance between two strings, which play an important role in a wide range of applications in different fields.

Consequently, this led us to develop a strong and effective method for this purpose. Although, developing highly accurate name matching algorithms is still a challenging issue in the research community (*Navarro, 2001*; *Hall & Dowling, 1980*). By deeply reviewing the previous studies, it found that several studies have been conducted to develop name-matching algorithms, which are used to cope with many important topics. The classification of these algorithms is implemented into two categories: approximate string matching (inexact) algorithms (*Al-Ssulami, 2015*; *Hall & Dowling, 1980*; *Navarro, 2001*) and exact

string-matching algorithms (*Al-Ssulami, 2015*; *Charras & Lecroq, 2004*; *Christen, 2006a*; *Christen, 2006b*).

Name identification and matching are increasingly used in several applications such as customer relation management (CRM), health care (HC), customer data integration (CDI), anti-money laundering (AML), criminal investigation (CI) and genealogy services (GS) (*Lisbach et al., 2013*). Besides, it is used also in other applications in the airports, plagiarism checking software, etc. If the matching is carried out considering only the exact similarity in such applications, it would be difficult and might be impossible to deal with the case of name variations, which is an unavoidable situation when dealing with real-world data sets (*Delgado et al., 2016*). That is, the exact matching approach is not suitable for large-scale applications and complex information systems, since it cannot retrieve names that have more than one acceptable spelling (*Christen, 2006a*; *Christen, 2006b*).

To have a highly effective name matching methods, the approximate string-matching approach should be adopted rather than exact matching. Therefore, this paper aims to develop an algorithm for name matching, that consider an approximate string-matching algorithm to allow dealing with possible technical or computational errors. Such matching algorithms have been used in several applications such as spelling correction (*Park et al., 2020*), linking databases (*Hand & Christen, 2018*), text retrieval (*Abdulhayoglu, Thijs & Jeuris, 2016*), handwriting recognition (*Chowdhury, Bhattacharya & Parui, 2013*), computational biology "DNA" (*Berger, Waterman & Yu, 2020*), and name recognition (*Delgado et al., 2016*), etc. Consequently, in this work, a new softened distance measure is proposed, based on the BI-DIST distance to increase the efficiency and accuracy of the name-matching method. This is achieved by identifying different cases that form bigram scales, grounded on statistical analysis to soften the distance scale. Accordingly, it is hypothesized that an evolutionary method can be adapted to adjust the weights of the distance scale between n-grams.

## BACKGROUND AND RELATED WORK

Many research works mainly concentrate on name matching methods improvement and algorithm complexity. In addition to the complex process of matching names as aforementioned, misspelling and different spelling of words are detected. The effective way is to apply an approximate string-matching technique to prevent the recurring of different spelling inputs and misspelling (*Lertnattee & Paluekpet, 2019*). Given two names X and Y represented as strings of n and m characters, respectively, the Edit Distance, aka Levenshtein Distance (LD), indicates the least possible cost of editing processes (insertion, deletion, and substitution) to convert X to Y (*Levenshtein, 1966*). For example, if X = "Zantac‖" and Y = "Xanax‖", the edit distance is 3 as the minimum transformation implies two substitution operations ("Z" → "X" and "c" → "x") and one deletion operation (letter "t"). Which is calculated using the recurrence formula in Eq. (1), The Levenshtein distance between two

strings s, t is given mathematically by $\text{Lev}_{s,t}(|s|, |t|)$ where.

$$\text{Lev}_{s,t}(i,j) = \begin{cases} \text{Max}(i,j) & if \ (\text{Min}(i,j) = 0) \\ \text{Min} \begin{cases} \text{Lev}_{s,t}(i,j-1)+1 \\ \text{Lev}_{s,t}(i-1,j)+1 \\ \text{Lev}_{s,t}(i-1,j-1)+1_{(si \neq tj)} \end{cases} & otherwise \ (1) \end{cases} \qquad (1)$$

In Eq. (1), 1 is the *indicator function* equal to 0 if $s_{i} = t_{j}$ and 1 otherwise. By $|s|$ we denote the length of the string s. $\text{Lev}_{s,t}(i,j)$ is the distance between string prefixes—the first i characters of s and the first j characters of t. The first part of this formula denotes the number of insertion or deletion steps to transform prefix into an empty string or vice versa. The second block is a recursive expression with the first line represents deletion and the second one represents insertion. The last line is responsible for substitutions. More details are available at (https://www.baeldung.com/cs/levenshtein-distance-computation).

In *Damerau (1964)*, Damerau–Levenshtein Distance (DLD) is presented which is akin to the LD algorithm. The chief modification is that DLD lets one more edit, particularly where the two adjacent characters can be transposed. The DLD algorithm describes the distance between two strings s and t by the following recursive relation as shown in Eq. (2):

$$\text{DLev}_{s,t}(i,j) = \text{Min} \begin{cases} 0 & if \ i = j = 0 \\ \text{DLev}_{s,t}(i-1,j)+1 & if \ i > 0 \\ \text{DLev}_{s,t}(i,j-1)+1 & if \ j > 0 \\ \text{DLev}_{s,t}(i-1,j-1)+1_{(si \neq tj)} & if \ i,j > 0 \\ \text{DLev}_{s,t}(i-2,j-2)+1 & if \ i,j > 1 \ and \ s[i] = t[j-1] \ and \ s[i-1] = t[j] \end{cases} \qquad (2)$$

Where $1_{(si \neq tj)}$ is the indicator function equal to 0 when $si = tj$ and equal to 1 otherwise.

In *Rees (2014)*, a customized approach called a Modified Damerau-Levenshtein Distance algorithm (MDLD) was proposed. MDLD was adjusted and tested against two input strings that support block transpositions of numerous characters. The MDLD algorithm's time complex $O(n^3)$, is presented algorithm (MDLD) in its Oracle PL/SQL form. More details are available at (https://confluence.csiro.au/public/taxamatch/the-mdld-modified-damerau-levenshtein-distance-algorithm).

The N-gram Distance (N-DIST) that was proposed by *Kondrak (2005)* in his research works by the fusion of features carried out by grams of size and non-crossing-links constraints, and the first letter is repeated initially. On the other hand, it is found that BI-DIST is a case of N-DIST (*Kondrak, 2005*). In *Abdulhayoglu, Thijs & Jeuris (2016)* each matrix element $NDIST_{s,t}(i,j)$ is calculated according to Eq. (3), where the cost in Eq. (4) is the total number of distinct letters in the same positions in the character n-grams $s_i; t_j$,

and n is the size of the character n-gram, as shown in Eqs. (3) and (4):

$$NDIST_{s,t}(i,j) = \begin{cases} Max(i,j) & (i=0 \text{ or } j=0) \\ Min \begin{cases} NDIST_{s,t}(i-1,j)+1 \\ NDIST_{s,t}(i,j-1)+1 \\ NDIST_{s,t}(i-1,j-1)+d_n\left(T_{i,j}^n\right). \end{cases} \end{cases} \tag{3}$$

$$d_n(T_{i,j}^n) = \frac{1}{n}\sum_{u=1}^{n} d_1(x_{i+u}, y_{j+u}), \tag{4}$$

*Kondrak (2005)* proposed the measures N-gram Distance and Similarity (N-DIST and N-SIM) respectively, where the recall metric is used to assess the results of twelve measures with the U.S. Pharmacopeia (USP) look-alike/sound-alike (LASA) list of 360 unique drug names. In this study, Kondrak concluded that combining BI-DIST and BI-SIM achieves the best results. The Food and Drug Administration (FDA) uses it to create automated warning systems to identify potential LASA errors in prescription electronic systems and phonetic orthographic computer analysis (POCA) software. Moreover, *Millán-Hernandez et al. (2019)* proposed a Soften Bigram Similarity measure (Soft-Bisim). This work concentrated on improving an algorithm to Identify Confusable Drug Names, based on Bi-gram algorithms and the blend of the longest common subsequences. Furthermore, the research work achieved (*Al-Hagree et al., 2019a*; *Al-Hagree et al., 2019b*) proposed an enhanced N-DIST method that concentrated on improving an algorithm for Name Matching. However, the previous studies differ from the contribution in this paper, because the proposed algorithm in this paper combines a Bi-gram technique with a distance technique (*Al-Hagree et al., 2019a*; *Al-Hagree et al., 2019b*).

## THE PROPOSED METHOD

In this section, a Soft-Bidist is presented. The Soft-Bidist measure is an extension of BI-DIST, it softening the scale of comparison among the name Bigrams for improving the name detection. This section organizes as follows. The first subsection is to describe the involved cases of bigrams in the scale of the Soft-Bidist distance. Then, the Minimum, Maximum, and Average functions, which are used as means to identify the weights in the distance scale by statistical means, are mentioned. It is thus assumed that an evolutionary approach identifies the best levels in the distance scale compared to the original distance scale that Kondrak proposed in BI-DIST (cf. Eqs. (3) and (4)). In other words, we consider this problem as an evolutionary approach for optimizing the internal parameters of the distance scale.

## Definition of Soft-Bidist distance

Let X and Y be given names represented as sequences of sizes n and m, respectively, Soft-Bidist is defined as follows:

$$BIDIST_{s,t}(i,j) = \begin{cases} Max(i,j) & (i=0 \text{ or } j=0) \\ Min \begin{cases} BIDIST_{s,t}(i-1,j) + ID_n\left(T_{i,j}^n\right). \\ BIDIST_{s,t}(i,j-1) + ID_n\left(T_{i,j}^n\right) \\ BIDIST_{s,t}(i-1,j-1) + d_n\left(T_{i,j}^n\right). \end{cases} \end{cases} . \quad (5)$$

The distance scale for Soft-Bidist is shown as follows:

$$d_n\left(T_{i,j}^n\right) = \begin{cases} wt_1, \text{ if}\left(S_{i-1} = T_{j-1}\right) \text{ and } \left(S_i = T_j\right) & \text{Case 1} \\ wt_2, \text{ if}\left(S_{i-1} \neq T_{j-1}\right) \text{ and } \left(S_i \neq T_j\right) & \text{Case 2} \\ \quad \text{and}\left(S_{i-1} \neq T_j\right) \text{ and } \left(S_i \neq T_{j-1}\right) \\ wt_3, \text{ if}\left(S_{i-1} = T_j\right) \text{ and } \left(S_i = T_{j-1}\right) & \text{Case 3} \\ wt_4, \text{ if}\left(S_{i-1} \neq T_{j-1}\right) \text{ and } \left(S_i = T_j\right) & \text{Case 4} \\ wt_5, \text{ if}\left(S_{i-1} = T_{j-1}\right) \text{ and } \left(S_i \neq T_j\right) & \text{Case 5} \\ wt_6, \text{ if}\left(S_{i-1} = T_j\right) \text{ and } \left(S_i \neq T_{j-1}\right) & \text{Case 6} \\ wt_7, \text{ if}\left(S_{i-1} \neq T_j\right) \text{ and } \left(S_i = T_{j-1}\right) & \text{Case 7} \end{cases} \quad (6)$$

$$ID_n\left(T_{i,j}^n\right) = \begin{cases} wt_8, \text{ if}(S_{i-1} = T_j) \text{ and } (S_i \neq T_{j-1}) & \text{Case 8} \\ wt_9, \text{ if}(S_{i-1} \neq T_j) \text{ and } (S_i = T_{j-1}) & \text{Case 9} \end{cases} \quad (7)$$

To increase the accuracy of identifying the names, there is a need to find the set of weights WT = {wt1; wt2; ...; wt9} of the distance scale of Soft-Bidist. For this, a randomized value is used (*Levenshtein, 1966*; *Al-Hagree et al., 2019a*; *Al-Hagree et al., 2019b*; *Kondrak, 2005*; *Millán-Hernandez et al. , 2019*; *Rees, 2014*).

## Finding the Distance Scale for Soft-Bidist

The cases are weighted as symbols. These weights are depend on Tables 1 and 2, which are used to adapt to the operational environment and get highly accurate results in various situations. Therefore, Table 1 contains several different weights. After changing the default values of [0, 1, 1, 0, 1, 1, 1, 1 and 1] with wt1, wt2, wt3, wt4, wt5, wt6, wt7, wt8, and wt9 for all cases respectively, the new weights achieve results similar to that obtained by LD algorithm. Again, other default values have been examined [0, 1, 0, 0, 1, 1, 1, 1 and 1] with for all cases respectively, the new weights achieve results similar to that obtained by the DLD algorithm. Finally, other default values of [0, 1, 1, 0.5, 0.5, 1, 1, 1 and 1] for wt1, wt2, wt3, wt4, wt5, wt6, wt7, wt8, and wt9 for all cases respectively, the new weights achieves results similar to that obtained by the N-DIST algorithm. Based on the previous weight values, new weights were added to Table 2.

## THE EXPERIMENTAL RESULTS

This section presents the experimental results that are carried out in this research. The objective of these experiments is to assess the Soft-Bidist algorithm compared with other

**Table 1** The various weights for Soft-Bidist that yelled similar results to other algorithms from the literature.

| Results similar to | Proposed Weights | | | | | | | | |
|---|---|---|---|---|---|---|---|---|---|
| | $wt_1$ | $wt_2$ | $wt_3$ | $wt_4$ | $wt_5$ | $wt_6$ | $wt_7$ | $wt_8$ | $wt_9$ |
| (LD). | 0, | 1, | 1, | 0, | 1, | 1, | 1, | 1, | 1, |
| (DLD). | 0, | 1, | 0, | 0, | 1, | 1, | 1, | 1, | 1, |
| N-DIST is $n = 2$ "BI" | 0, | 1, | 1, | 0.5, | 0.5, | 1, | 1, | 1, | 1, |

**Table 2** The randomize of weights for Soft-Bidist algorithm.

| No | Weights for Soft-Bidist | $wt_1,$ | $wt_2,$ | $wt_3,$ | $wt_4,$ | $wt_5,$ | $wt_6,$ | $wt_7,$ | $wt_8$ | $wt_9$ |
|---|---|---|---|---|---|---|---|---|---|---|
| 1 | Minimum. | 0 | 1 | 0 | 0, | 0.5, | 1, | 1, | 1 | 1 |
| 2 | Average. | 0 | 1 | 0.7, | 0.2, | 0.8, | 1, | 1, | 1 | 1 |
| 3 | Maximum (Cases 8, 9 is 0.5). | 0 | 1 | 0, | 0, | 0.5, | 1, | 1, | 0.5 | 0.5 |
| 4 | Average (Cases 8, 9 is 0.5). | 0 | 1 | 0.7, | 0.2, | 0.8, | 1, | 1, | 0.5 | 0.5 |
| 5 | | 0 | 1 | 1, | 0, | 0.5, | 1, | 1, | 1 | 1 |
| 6 | | 0, | 1, | 0, | 0, | 0.5, | 1, | 1, | 1 | 1 |
| 7 | | 0, | 1, | 0, | 0.2, | 0.2, | 1, | 1, | 1 | 1 |
| 8 | | 0, | 1, | 0, | 0.1, | 0.1, | 1, | 1, | 1 | 1 |
| 9 | Randomize | 0, | 1, | 0, | 0, | 0.2, | 1, | 1, | 1 | 1 |
| 10 | weights | 0, | 1, | 1, | 0, | 0.5, | 1, | 1, | 0.5 | 0.5 |
| 11 | | 0, | 1, | 0.5, | 0, | 0.5, | 1, | 1, | 0.5 | 0.5 |
| 12 | | 0, | 1, | 0, | 0.1, | 0.1, | 1, | 1, | 0.5 | 0.5 |
| 13 | | 0, | 1, | 0.5, | 0, | 0.1, | 1, | 1, | 0.5 | 0.5 |
| 14 | | 0, | 1, | 0, | 0, | 0.2, | 1, | 1, | 0.5 | 0.5 |
| 15 | | 0, | 1, | 0, | 0, | 0.1, | 1, | 1, | 0.5 | 0.5 |
| 16 | The applied in this paper. | 0, | 1, | 0, | 0.2, | 0.2, | 1, | 1, | 0.5 | 0.5 |

algorithms from the literature. Due to the absence of standard datasets for name matching, different multilingual datasets (English, Arabic, Portuguese) is used in the experiments carried out in this research. These datasets are presented by (*Al-Hagree et al., 2019a*; *Al-Hagree et al., 2019b*; *Ahmed & Nürnberger, 2009*; *Rees, 2014*; *Al-Sanabani & Al-Hagree, 2015*). Different spelling errors and typographical are included in these datasets. In our previous work, a modified algorithm was applied to drug names in English documents, but for current work, the Soft-Bidist is applied to the different datasets deals with personal names in Arabic, English and Portuguese. To our knowledge, there have been no previous reports of good performance on this combination of datasets with different languages.

The same data preparation used in *Al-Hagree et al. (2019a)*; *Al-Hagree et al. (2019b)*) is used in this research to be compared with the Soft-Bidist algorithm.

After defining the default values of [0, 1, 1, 0.2, 0.2, 1, 1, 1 and 1] for for all cases respectively, the proposed algorithm appears to have achieved high accurate results. Generally, it is not easy to provide accurate weights between pair source and target. In order to gain proper values for weights wt1, wt2, wt3, wt4, wt5, wt6, wt7, wt8, and wt9, the

Table 3 The results with different weights for Soft-Bidist.

| No | $wt_1, wt_2, wt_3, wt_4, wt_5, wt_6, wt_7, wt_8$ and $wt_9$ | Average (percentage similarity) |
|---|---|---|
| 1 | 0,1,1,0,1,1,1,1 and 1 | 0.83 |
| 2 | 0,1,0,0,1,1,1,1 and 1 | 0.87 |
| 3 | 0,1,1,0.5,0.5,1,1,1 and 1 | 0.82 |
| 4 | 0,1,0,0.5,0.5,1,1,1 and 1 | 0.89 |
| 5 | 0,1,1,0,0.5,1,1,1 and 1 | 0.87 |
| 6 | 0,1,0,0,0.5,1,1,1 and 1 | 0.91 |
| 7 | 0,1,0,0.2,0.2,1,1,1 and 1 | 0.91 |
| 8 | 0,1,0,0.1,0.1,1,1,1 and 1 | 0.93 |
| 9 | 0,1,0,0,0.2,1,1,1 and 1 | 0.93 |
| 10 | 0,1,1,0,0.5,1,1,0.5 and 0.5 | 0.89 |
| 11 | 0,1,0.5,0,0.5,1,1,0.5 and 0.5 | 0.91 |
| 12 | 0,1,0,0,0.5,1,1,0.5 and 0.5 | 0.93 |
| 13 | 0,1,0.5,0,0.5,1,1,0.5 and 0.5 | 0.88 |
| 14 | 0,1,0,0.2,0.2,1,10.5 and 0.5 | **0.94** |
| 15 | 0,1,0,0.1,0.1,1,1,0.5 and 0.5 | 0.96 |
| 16 | 0,1,0.5,0,0.,1,1,0.5 and 0.5 | 0.95 |
| 17 | 0,1,0,0,0.2,1,1,0.5 and 0.5 | 0.96 |
| 18 | 0,1,0,0,0.1,1,1,0.5 and 0.5 | 0.97 |

experiments with different weights for Tables 1 and 2 of dataset 1 (*Al-Hagree et al., 2019a*; *Al-Hagree et al., 2019b*) should be repeated. The results are presented in Table 3.

The experiments are repeated on dataset 2 (*Ahmed & Nürnberger, 2009*; *Al-Sanabani & Al-Hagree, 2015*) for the Soft-Bidist algorithms. Table 4 shows the result of this experiment. It can be noticed that the Soft-Bidist Algorithm functions better than the DLD, LD and N-DIST algorithms particularly being compared with names transposition such as the names that are shown in rows 3 and 4. Unlike DLD, LD, and N-DIST, the Soft-Bidist algorithm is sensitive to replacement as shown in rows 6 and 7. The Soft-Bidist Algorithm computes recurring letters, detection of errors, and deletion in a more proficient manner than DLD, LD, and N-DIST as they appear in rows 5, 8, 9, 10, 11, 12, 13, and 14. The Soft-Bidist algorithm exhibits a number of advantages over the DLD, LD, and N-DIST algorithms as aforementioned. Therefore, the Soft-Bidist algorithm functions well and gives a better accuracy compared with the DLD, LD, and N-DIST algorithms for all pairs in dataset 2 as appears in Table 4.

Furthermore, more experiments are implemented with various datasets to prove the evidence of the ability of the Soft-Bidist algorithm. Ten datasets are chosen and implemented on the DLD, LD, N-DIST, MDLD, and Soft-Bidist algorithms as appears in Table 5. That demonstrates the evidence and ability of the Soft-Bidist algorithm in name matching.

In Table 5, the Soft-Bidist algorithm gets 93% and 90% while DLD, LD, N-DIST, and MDLD algorithms get 88%, 88%, 86%, and 89%, respectively. Therefore, the Soft-Bidist algorithm gives more accurate results than the DLD, LD, N-DIST, and MDLD algorithms

**Table 4  Comparison between algorithm from literature and Soft-Bidist.**

| No. | String | | Proposed Algorithms | | | | |
|---|---|---|---|---|---|---|---|
| | | | DLD | LD | N-DIST | Soft-Bidist (0,1,0,0.5,0.5,1,1,1 and 1) | Soft-Bidist (0,1,0,0.2,0.2,1,1,0.5 and 0.5) |
| | Source | Target | Sim % | Sim % | Sim % | Sim % | Sim % |
| 1 | precede | preceed | 0.86 | 0.71 | 0.79 | 1.00 | 0.97 |
| 2 | promise | promiss | 0.86 | 0.86 | 0.93 | 1.00 | 0.97 |
| 3 | absence | absense | 0.86 | 0.86 | 0.86 | 0.86 | 0.94 |
| 4 | achieve | acheive | 0.86 | 0.71 | 0.71 | 0.86 | 0.94 |
| 5 | accidentally | accidentaly | 0.92 | 0.92 | 0.92 | 0.96 | 0.96 |
| 6 | algorithm | algorythm | 0.89 | 0.89 | 0.89 | 0.89 | 0.96 |
| 7 | similar | Similer | 0.86 | 0.86 | 0.86 | 0.86 | 0.94 |
| 8 | dilemma | Dilemma | 0.86 | 0.86 | 0.86 | 0.93 | 0.93 |
| 9 | almost | allmost | 0.86 | 0.86 | 0.86 | 0.93 | 0.93 |
| 10 | amend | ammend | 0.83 | 0.83 | 0.83 | 0.92 | 0.92 |
| 11 | occurred | occured | 0.88 | 0.88 | 0.88 | 0.94 | 0.94 |
| 12 | embarrass | embarass | 0.89 | 0.89 | 0.89 | 1.00 | 0.94 |
| 13 | harass | harrass | 0.86 | 0.86 | 0.86 | 1.00 | 0.93 |
| 14 | really | Realy | 0.83 | 0.83 | 0.83 | 0.92 | 0.92 |
| | Average (percentage similarity) | | 0.86 | 0.84 | 0.86 | 0.84 | **0.94** |

**Note.**
The best results are shown in bold.

for all datasets, because LD, DLD, N-DIST, and MDLD algorithms have not considered the transposition operations of Latin-based language especially the English language.

## COMPARATIVE STUDY FOR SOFT-BIDIST ALGORITHM AND COMPARED ALGORITHMS

The dataset used for comparison in this section has been extracted manually from the book of (*Christen, 2012*). To clarify the way that string comparison functions approximate various similarity estimations when used for similar strings. Table 6 gives sample results when given names and surnames are compared for the Soft-Bidist algorithm and compared algorithms as well. The highest similarity is shown in bold, while the lowest is shown in italics. The similarity values in Table 6 are calculated based on chosen name pairs. Table 6 reflects how different string comparison functions produce various similarity approximates for the same name pairs. According to the given results, there are significant differences in the similarities approximated on the same pair. These functions have various characteristics concerning the average and the spread of the value of similarity. Methods as Winkler, Jaro, the compression-based comparison operates, and Soft-Bidist Algorithm gives the highest mean of similarity values. Whereas, the edit distance (ED), the longest common substring (LCS) comparison, and the q-gram ('n-gram') based functions (*Ukkonen, 1992*) result in a much lower mean in the similarity values as can be seen in Table 6.

Hadwan et al. (2021), *PeerJ Comput. Sci.*, DOI 10.7717/peerj-cs.465

**Table 5  The mean similarity of LD, DLD, N-DIST, MDLD and Soft-Bidist algorithms with different datasets.**

| | Datasets | DLD (0,1,0,0,1, 1,1,1 and 1) Sim % | LD (0,1,1,0,1, 1,1,1 and 1) Sim % | N-DIST (0,1,1,0.5, 0.5,1,1,1 and 1) Sim % | MDLD Sim % | Soft-Bidist (0,1,0,0.5,0.5, 1,1,1 and 1) Sim % | Soft-Bidist (0,1,0,0.2,0.2, 1,1,0.5 and 0.5) Sim % |
|---|---|---|---|---|---|---|---|
| 1 | Dataset 3 (English 60 pairs) (*Al-Hagree et al., 2019a*). | 0.86 | 0.83 | 0.81 | 0.87 | 0.89 | **0.94** |
| 2 | Dataset 4 (English 4013 pairs) (*Al-Hagree et al., 2019a*). | 0.85 | 0.84 | 0.82 | 0.86 | 0.89 | **0.92** |
| 3 | Dataset5 (Portuguese 120 pairs) (*Ahmed & Nürnberger, 2009*). | 0.84 | 0.84 | 0.82 | 0.84 | 0.85 | **0.91** |
| 4 | Dataset 6 'CAAB' (641 pairs) (*Rees, 2014*). | 0.94 | 0.95 | 0.93 | 0.94 | 0.95 | **0.96** |
| 5 | Dataset 7 'Dalcin name pairs' (171 pairs) (*Rees, 2014*). | 0.95 | 0.94 | 0.93 | 0.95 | 0.97 | 0.97 |
| 6 | Dataset 8 'CAABWEB' (2047 pairs) (*Rees, 2014*)]. | 0.93 | 0.93 | 0.92 | 0.93 | 0.95 | **0.95** |
| 7 | Dataset 9 'GRIN genera' (189 pairs) (*Rees, 2014*). | 0.89 | 0.88 | 0.87 | 0.89 | 0.90 | **0.94** |
| 8 | Dataset 10 'CAAB Genera' (115 pairs) (*Rees, 2014*). | 0.90 | 0.88 | 0.85 | 0.90 | 0.91 | **0.94** |
| 9 | Dataset 11 'CAABWEB Genera' (853 pairs) (*Rees, 2014*). | 0.88 | 0.88 | 0.87 | 0.88 | 0.90 | **0.93** |
| 10 | Dataset 12 'Arabic name (600 pairs) (*Al-Sanabani & Al-Hagree, 2015*). | 0.80 | 0.79 | 0.73 | 0.80 | 0.77 | 0.80 |
| | Similarity mean | 0.88 | 0.88 | _0.86_ | 0.89 | **0.90** | **0.93** |

**Note.**
The best results are shown in bold, and the worst results are underlined.

**Table 6  The average similarities for proposed weights and compared methods presented at *Christen (2012)*.**

| No. | Algorithms | Average similarity |
|---|---|---|
| 1 | Jaro | 0.86 |
| 2 | Winkler | **0.88** |
| 3 | Bigram | 0.62 |
| 4 | Trigram | *0.52* |
| 5 | Positional bigrams | 0.62 |
| 6 | Skip-grams | 0.62 |
| 7 | Levenshtein edit distance (LD) | 0.70 |
| 8 | Damerau-Levenshtein edit distance (DLD) | 0.72 |
| 9 | BagDist | 0.78 |
| 10 | Editex | 0.75 |
| 11 | compression-based similarity using the ZLib compressor | 0.66 |
| 12 | longest common substring (length = 2) | 0.67 |
| 13 | longest common substring (length = 3) | 0.60 |
| 14 | Smith-Waterman edit distance | 0.65 |
| 15 | syllable alignment distance | 0.66 |
| 16 | MDLD | 0.72 |
| 17 | 0,1,1,0,1,1,1,1 and 1(Sof-Bidist) | 0.70 |
| 18 | 0,1,0,0,1,1,1,1 and 1 (Sof-Bidist) | 0.72 |
| 19 | 0,1,1,0.5,0.5,1,1,1 and 1 (Sof-Bidist) | 0.68 |
| 20 | 0,1,0,0.5,0.5,1,1,1 and 1 (Sof-Bidist) | 0.77 |
| 21 | 0,1,0,0.2,0.2,1,1,1 and 1 (Sof-Bidist) | 0.78 |
| 22 | 0,1,0,0.1,0.1, 0.5, 0.5, 0.5 and 0.5 (Sof-Bidist) | 0.83 |
| 23 | 0,1,0,0,0.1, 0.5, 0.5, 0.5 and 0.5 (Sof-Bidist) | 0.85 |
| 24 | 0,1,0,0,0.1, 0.2, 0.2, 0.2 and 0.2 (Sof-Bidist) | **0.88** |

**Note.**
The best results are shown in bold.

**Table 7  Correspondence between the predicted and the actual classes.**

| Algorithm | | Predicted | |
|---|---|---|---|
| | | **Match** | **Not Match** |
| Actual (Truth) | Match | True Positive (TP) | False Negative (FN) |
| | Not Match | False Positive (FP) | True Negative (TN) |

## The estimated measure

The estimated measure is using the f-measure which is also called f-score. The name matching quality has proven to be effective (*Christen, 2006a*; *Christen, 2006b*; *Kolomvatsos et al., 2013*), which is based on precision and recall. These metrics are used for classification tasks. They compare the predicted class of an item with the actual class, as shown in Table 7. Based on Table 7 and following (Kolomvatsos et al., 2013), precision and recall are defined as:

**Table 8** The results of average F-measure values.

| | Datasets | Compared Algorithm | | | | Proposed Algorithm | |
|---|---|---|---|---|---|---|---|
| | | LD | DLD | N-DIST | MDLD | Soft-Bidist (0,1,0,0.5,0.5,1,1,1 and 1) | Soft-Bidist (0,1,0,0.2,0.2,1,1,0.5 and 0.5) |
| | | Sim % | Sim % | Sim % | Sim % | Sim % | Sim % |
| 1 | Dataset 3 (English 60 pairs) | 0.77 | 0.85 | 0.74 | 0.85 | 0.91 | **0.95** |
| 2 | Dataset 4 (English 4013 pairs) | 0.75 | 0.76 | 0.73 | 0.89 | 0.90 | **0.94** |
| 3 | Dataset 5 (Portuguese 120 pairs) | 0.80 | 0.80 | 0.77 | 0.80 | 0.82 | **0.93** |
| 4 | Dataset 6 'CAAB' (641 pairs) | 0.99 | 0.99 | 0.99 | 0.99 | **1.00** | **1.00** |
| 5 | Dataset 7 'Dalcin name pairs' (171 pairs) | 1.00 | 1.00 | 1.00 | 1.00 | **1.00** | **1.00** |
| 6 | Dataset 8 'CAABWEB' (2047 pairs) | 0.96 | 0.97 | 0.94 | 0.98 | 0.98 | **0.99** |
| 7 | Dataset 9 'GRIN genera' (189 pairs) | 0.93 | 0.94 | 0.88 | 0.94 | **0.95** | **0.95** |
| 8 | Dataset 10 'CAAB Genera' (115 pairs) | 0.95 | 0.96 | 0.88 | 0.96 | **0.97** | 0.96 |
| 9 | Dataset 11 'CAABWEB Genera' (853 pairs) | 0.91 | 0.93 | 0.84 | 0.79 | 0.91 | **0.94** |
| 10 | Dataset 1 'Arabic name (600 pairs) | 0.66 | 0.68 | 0.53 | 0.68 | 0.70 | **0.77** |
| | F-MEASURE MEAN | 0.87 | 0.89 | 0.83 | 0.89 | 0.91 | **0.94** |

**Note.**
The best results are shown in bold, and the worst results are underlined.

$$\text{Precision} = \frac{\text{TP}}{\text{TP} + \text{FP}} \tag{8}$$

$$\text{Recall} = \frac{\text{TP}}{\text{TP} + \text{FN}} \tag{9}$$

Moreover, the F-measure is defined as the weighted combination of precision and recall. The F-measure is defined by:

$$\text{F-Measure} = \frac{2.\text{Precision}.\ \text{Recall}}{\text{Precision} + \ \text{Recall}} \tag{10}$$

Since F-measure is an accuracy measure between 0 and 1, the higher the values, the better and more accurate are the results. The experiments can be seen in Table 8, the mean of f-measures achieved by the proposed Soft-Bidist algorithm on all instances for the used dataset and the threshold is 0.94, which outperforms the other algorithms. Best results shown boldface and worst results underlined. The thresholds are 0.90, 0.85, 0.80, 0.75, 0.70 and 0.65 of all datasets tested (three English datasets, one Portuguese dataset, three species datasets, three genera datasets, and one Arabic dataset).

Table 9 presents the F1-scores for different scenarios. For the dataset 5 (Portuguese 120 pairs), using different Edit Distance. The best results were retrieved with the threshold values for a correct match of 0.65, 0.70, 0.75, 0.80, 0.85 and 0.90 for LD, DLD, N-DIST, MDLD and Soft-Bidist, respectively (*Abdulhayoglu, Thijs & Jeuris, 2016*). Table 9 shows F-measure vs. Threshold curves for dataset 5 (Portuguese 120 pairs).

Repeating the previous experiment has been carried based on all Datasets Table 10. The proposed algorithms Soft-Bidist (0,1,0,0.2,0.2,1,1,1 and 1) and Soft-Bidist

**Table 9  F1-scores of different algorithms, thresholds and similarity calculation.**

| | | Thresholds | | | | | |
|---|---|---|---|---|---|---|---|
| | **Algorithms** | **65** | **70** | **75** | **80** | **85** | **90** |
| 1 | LD | 0.987 | 0.961 | 0.938 | 0.889 | 0.750 | 0.273 |
| 2 | DLD | 0.987 | 0.961 | 0.938 | 0.894 | 0.750 | 0.273 |
| **3** | N-DIST | 0.966 | 0.952 | 0.924 | 0.863 | 0.710 | 0.222 |
| 4 | MDLD | 0.987 | 0.961 | 0.938 | 0.894 | 0.750 | 0.273 |
| **5** | 0,1,0,0.2,0.2,1,1,1 and 1 (Soft-Bidist) | **0.987** | **0.970** | **0.966** | **0.938** | **0.909** | **0.794** |

**Note.**
The best results are shown in bold.

**Table 10  The results of F-measure mean values.**

| | | Compared Algorithm | | | | Proposed Algorithm | |
|---|---|---|---|---|---|---|---|
| | **Datasets** | **LD** | **DLD** | **N-DIST** | **MDLD** | **Soft-Bidist (0,1,0,0.2,0.2,1, 1,0.5 and 0.5)** | **Soft-Bidist (0,1,0,0.2,0.2, 1,1,0.5 and 0.5)** |
| | | Sim % | Sim % | Sim % | Sim % | Sim % | Sim % |
| 1 | Dataset 3 (English 60 pairs) | 0.77 | 0.85 | _0.74_ | 0.85 | **0.95** | **1.00** |
| 2 | Dataset 4 (English 4013 pairs) | 0.75 | 0.76 | _0.73_ | 0.89 | **0.94** | **0.99** |
| 3 | Dataset 5 (Portuguese 120 pairs) | _0.80_ | _0.80_ | _0.77_ | _0.80_ | **0.93** | **0.95** |
| 4 | Dataset 6 'CAAB' (641 pairs) | 0.99 | 0.99 | 0.99 | 0.99 | **1.00** | **1.00** |
| 5 | Dataset 7 'Dalcin name pairs' (171 pairs) | 1.00 | 1.00 | 1.00 | 1.00 | **1.00** | **1.00** |
| 6 | Dataset 8 'CAABWEB' (2047 pairs) | 0.96 | 0.97 | 0.94 | 0.98 | **0.99** | **0.99** |
| 7 | Dataset 9 'GRIN genera' (189 pairs) | 0.93 | 0.94 | _0.88_ | 0.94 | **0.95** | **0.99** |
| 8 | Dataset 10 'CAAB Genera' (115 pairs) | 0.95 | 0.96 | _0.88_ | 0.96 | **0.96** | **0.99** |
| 9 | Dataset 11 'CAABWEB Genera' (853 pairs) | 0.91 | 0.93 | _0.84_ | 0.79 | **0.94** | **0.98** |
| 10 | Dataset 1 'Arabic name (600 pairs) | 0.66 | 0.68 | _0.53_ | 0.68 | **0.77** | **0.81** |
| | F-MEASURE MEAN | 0.87 | 0.89 | _0.83_ | 0.89 | **0.94** | **0.97** |

**Note.**
The best results are shown in bold, and the worst results are underlined.

(0,1,0,0.2,0.2,1,1,0.5 and 0.5) gives more accurate results than the algorithms LD,DLD,N-DIS and MDLD for all datasets as shown in Table 10. The mean of f-measures on all datasets as can be seen in Table 10, which equals 0.97, says that accuracy is almost high and reasonable to trust the results. The best results are shown in bold, and the worst results are underlined. The thresholds are 0.90, 0.85, 0.80, 0.75, 0.70 and 0.65 of all datasets tested (three English datasets, one Portuguese dataset, three species datasets, three genera datasets, and one Arabic dataset).

## CONCLUSION

In this research, Soft-Bidist is proposed where it used a new methodology for improving name-matching accuracy. The Soft-Bidist algorithm handles the transposition, deletion, substitution, and insertion operations in a new way. These operations are dealt with differently, considering its different states of the name matching to enhance the matching

performance. Furthermore, different weights were assigned for each operation, which in turn enhanced the whole matching process. In comparison with other algorithms from the literature, the results of the experiments prove that the Soft-Bidist outperformed compared algorithms significantly. For future studies, it is suggested to explore the evolutionary algorithms to get the most proper weights for the soft calculation case, genetic algorithm (GA) for instance.

### Funding
The publication funding for this project was provided by the Deanship of Scientific Research, Qassim University. The funders had no role in study design, data collection and analysis, decision to publish, or preparation of the manuscript.

### Grant Disclosures
The following grant information was disclosed by the authors:
Deanship of Scientific Research, Qassim University.

### Competing Interests
The authors declare there are no competing interests.

### Author Contributions
- Mohammed Hadwan and Mohammed A. Al-Hagery conceived and designed the experiments, performed the experiments, analyzed the data, performed the computation work, prepared figures and/or tables, authored or reviewed drafts of the paper, and approved the final draft.
- Maher Al-Sanabani conceived and designed the experiments, performed the computation work, prepared figures and/or tables, authored or reviewed drafts of the paper, and approved the final draft.
- Salah Al-Hagree conceived and designed the experiments, performed the experiments, analyzed the data, performed the computation work, prepared figures and/or tables, and approved the final draft.

### Data Availability
The code is available at GitHub: Available at https://github.com/salahalhagree/Soft-Bigram-Distance. The code is written in C# 2008 version.

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
