# Peer review of "Soft Bigram distance for names matching"

_PeerJ Computer Science, doi:10.7717/peerj-cs.465_

## Round 0.1 · original submission · Major Revisions

Pay a special attention to a suggestion of the reviewer 2 that the manuscript "needs a complete rewrite before it is ready for publication". Provide full descriptions of your databases and hypotheses.

·

Basic reporting

The aligment of the formulas must be improved.

Soft-bisim is not described

Experimental design

no comment

Validity of the findings

In related work, drug names are used to explain some names-matching measures. However, the Soft-Bisim measure is not used in the experimentation.

The dataset is interesting because it has different languages and domains, but the experimentation and conclusion do not provide more information about it.

·

Basic reporting

This paper is exceptionally difficult to understand.

The recurrence relation at 78 differs, for no obvious reason, from the standard one found in most literature, This should be:

Lev(0, 0) = 0.0
Lev(i, 0) = Lev(i-1,0) + w['del'] when i > 0
Lev(0, j). = Lev(0,j-1) + w['ins'] when j > 0
Lev(i, j) = MIN(Lev(i-1, j-1] + (w['sub'] if x != y else 0),
Lev(i-1, j) + w['del'],
Lev[i ,j-1[ + w['ins'] ) when i > 0 and j > 0

In other words, the base cases of the first cell, the rest of the first column and the first row need to be clearly defined, and the main minimization needs to be defined with precision.

The definition at 86 is similar, and the authors correctly state that the only difference is the addition of an extra path to the minimization, to allow transposition to have a different score from the combination of an insertion and a deletion.

The mentions of Crt (not in the formula, but introduced at 99) are mysterious. Not sure what is meant here. The sentence starting at 90 is hard to understand: I can't make out what the N is in the claim that something is O(N^3) and the reference to an Oracle database is extremely unexpected.

I can't see the relationship between the formulae at 100 and 101 to the much clearer algorithms and presentation in Kondrak 2005 (which I accessed via semantic scholar at

@inproceedings{Kondrak2005NGramSA,
title={N-Gram Similarity and Distance},
author={Grzegorz Kondrak},
booktitle={SPIRE},
year={2005}
})

In particular T_{i,j}^n is undefined.

The paper needs to provide a more complete and self-contained explanation of what the proposed algorithm is. Even after re-reading the literature references, I am still not sure, and I think other readers will have the same problem.

Examining the code published on Github did not help me.

If asked to guess, what I think is happening is that the authors are adjusting some of the weights in the definition of BI-DIST and combining it with some distance measure.

The language of the paper is mostly professional, but leaves many important details unstated.

The literature references are appropriate. The summaries of what is in the literature are mostly accurate, but again frequently unclear.

The hypotheses of the paper are not clearly stated, because the experiments are not clearly described in the paper. It is possible that literature references may contain details of data preparation and experimental hypotheses, but this should not be necessary. The paper should contain sufficient detail that the reader can understand what the hypotheses are and against what data they are being tested. This material should be comprehensible even before the reader goes to the related literature. As written, the paper is insufficiently self-contained

The formal presentation of algorithms is confusing, with undefined symbols and unfamiliar notation. The authors are urged to go back to the literature that they rely on, then use notations and algorithm presentations that are more clearly similar to those in the prior work.

Experimental design

This is original primary research in line with the goals of the journal.

Experiments are reported at 153 and following. It is not clear from the paper which datasets are used and what metrics are being reported. This should be more self-contained. The authors need to include brief descriptions of the datasets used, the metrics that are used to evaluate performance and the way in which the reports of performance relate to identifiable knowledge gaps. For example, such a description could say "In previous work, a similar algorithm was applied to drug names in English documents, but our new dataset deals with personal names in Urdu. To our knowledge, there have been no previous reports of good performance on this combination of language and task. Our system achieves a balanced F1 of 0.93,"

The investigation is clearly performed to a high ethical standard. The technical quality of the investigation is unclear, because insufficient details are given on the datasets, the partitioning of the data into training, development and test partitions, or of the precise algorithms used.

The unclarity of the paper means that the results will be difficult or impossible to reproduce. Sadly, the source code on github is not much help here, because it is mostly undocumented.

Validity of the findings

The discussion of the findings is relatively good, with an intriguing suggestion for further work at 227

The underlying data has not been presented in an orderly fashion. Robustness, statistical soundness and appropriate controls are difficult to assess. The claims of superiority in 153-218 are too vague, and need to be supported by clearly reported tests of significant difference. The summary statement of improvement at 225-226 is not warranted by any clear evidence in the paper.

Additional comments

Overall, this paper reports work that could be interesting, but the presentation is so unclear that it needs a complete rewrite before it is ready for publication.

I believe that the idea of tuning the weights in a suitable string distance or string similarity function is fundamentally sound, but the suggestion at 63-65 that the work already uses an evolutionary algorithm seems to conflict with the rest of the paper, where what is described is an informal search for a good set of weights. It is fine to say (at 227) that a GA might help, but on its own the comment at 63-65 seems either premature or insufficiently supported by the remainder of the paper.

---

## Round 0.2 · accepted · Accept

Thank you for addressing all the concerns that were raised by our reviewers.

·

Basic reporting

The authors have addressed all the concerns that I raised in my previous review. The paper is now comprehensible and explains what was done and why.

Experimental design

Technically and ethically sound. The impediments to replicability are gone.

Validity of the findings

The results are trustworthy.